# The Urogenital System’s Role in Diseases: A Synopsis

**DOI:** 10.3390/cancers14143328

**Published:** 2022-07-08

**Authors:** Maroun Bou Zerdan, Rita Moukarzel, Nour Sabiha Naji, Yara Bilen, Arun Nagarajan

**Affiliations:** 1Department of Internal Medicine, SUNY Upstate Medical University, Syracuse, NY 13210, USA; bouzerdm@upstate.edu; 2Department of Hematology and Oncology, Cleveland Clinic Florida, Weston, FL 33331, USA; 3Faculty of Medicine, Lebanese American University Medical Center, Lebanese American University, Beirut 1102, Lebanon; rita.moukarzel01@lau.edu; 4Faculty of Medicine, American University of Beirut, Beirut 2020, Lebanon; nsn09@aub.edu.lb; 5Department of Internal Medicine, Cleveland Clinic Foundation, Cleveland, OH 44195, USA; bileny@ccf.org

**Keywords:** microbiota, urogenital system, renal cell carcinoma, immune checkpoint inhibitor

## Abstract

**Simple Summary:**

The urinary tract microbiome has come under a lot of scrutiny, and this has led to the rejection of the pre-established concept of sterility in the urinary bladder. Microbial communities in the urinary tract have been implicated in the maintenance of health. Thus, alterations in their composition have also been associated with different urinary pathologies, such as urinary tract infections. For that reason, tackling the urinary microbiome of healthy individuals, as well as its involvement in disease through the proliferation of opportunistic pathogens, could open a potential field of study, leading to new insights into prevention, diagnosis, and treatment strategies for different diseases.

**Abstract:**

The human microbiota contains ten times more microbial cells than human cells contained by the human body, constituting a larger genetic material than the human genome itself. Emerging studies have shown that these microorganisms represent a critical determinant in human health and disease, and the use of probiotic products as potential therapeutic interventions to modulate homeostasis and treat disease is being explored. The gut is a niche for the largest proportion of the human microbiota with myriad studies suggesting a strong link between the gut microbiota composition and disease development throughout the body. More specifically, there is mounting evidence on the relevance of gut microbiota dysbiosis in the development of urinary tract disease including urinary tract infections (UTIs), chronic kidney disease, and kidney stones. Fewer emerging reports, however, are suggesting that the urinary tract, which has long been considered ‘sterile’, also houses its unique microbiota that might have an important role in urologic health and disease. The implications of this new paradigm could potentially change the therapeutic perspective in urological disease.

## 1. Introduction

Renal cell carcinoma (RCC) has long been considered a rare entity; however, its incidence, morbidity, and mortality continue to increase annually to the present day, with an estimated 76,000 new cases in the United States in 2021 [1]. RCC is the most common type of malignant kidney neoplasm and one of the most aggressive cancers worldwide [2]. The treatment of choice for localized RCC has been surgical resection because it has a poor response to conventional chemotherapy and radiation therapy [3]. However, around 33% of patients are initially diagnosed with metastatic RCC (mRCC) at presentation, and around 50% of those diagnosed with localized disease relapse post-surgery [4]. Metastatic RCC has a very poor prognosis until today, and so its management and clinical outcome prediction remain a medical challenge [5].

The introduction of intrinsic therapies targeting specific signaling pathways has improved patient progression-free survival (PFS) [2]. Over the years, the treatment paradigm for metastatic RCC has evolved from cytokine-based therapies to vascular endothelial growth factor-targeting agents, to mTOR-inhibitors, tyrosine kinase inhibitors, and most recently, immune checkpoint inhibitors (ICIs) have become the mainstay treatment [6,7]. Targeting the programmed cell death-1 protein (PD-1), programmed cell death-ligand 1 (PD-L1), and cytotoxic T lymphocyte-associated antigen 4 (CTLA-4), ICIs have truly revolutionized metastatic RCC management [8], and their use in different combination strategies has become a novel standard of care in this patient population [9]. Nevertheless, despite the remarkable success in slowing down tumorigenesis and metastasis owing to the current era of targeted ICI therapy, clinical outcomes remain unsatisfactory, and survival prognosis in mRCC remains low [9]. Legitimate concerns remain relating in part to the inevitable resistance developed during ICI treatment [2] and also to the treatment-related immune adverse events and toxicities [10]. Moreover, there exists a great unpredictable variability in the efficacy and clinical response to ICI therapy across individual mRCC patients [11]. Hence, there is a critical need for reliable biomarkers and predictors for drug resistance, adverse events, and differential clinical response in mRCC patients treated with ICIs. Moreover, the introduction of alternative or adjunctive interventions to improve patient outcomes is another valid question to tackle.

## 2. Gut Microbiota’s Role in RCC Oncogenesis and Immunotherapy

The gut microbiota represents a vast collection of unique bacterial strains and microorganisms residing in the gastrointestinal tract and shaping a key element of host immunity that regulates oncogenesis, response to immunotherapy, and drug efficacy, resistance, and toxicity [12]. First, it has been widely reported in multiple meta-analyses that antibiotics (AB), by modifying the gut microbiota composition, cause gut dysbiosis and consequently poorer response to ICIs [9,10]. Ueda et al. [8] demonstrated, in a retrospective study, reduced objective response rate and shorter PFS for patients on ICIs treated with AB within 30 days prior compared with those that did not receive AB, presenting a negative association between AB and ICI treatment outcome. In their prospective study, Derosa et al. [10] reported analogous conclusions on mRCC and non-small-cell lung cancer patients treated with anti-PD-L1 targeted therapy, and they proved that AB induce gut dysbiosis such that they increase bacteria associated with resistance to ICIs, namely *Clostridium hathewayi* and *Clostridium clostridioforme*. Hence, they concluded that there are profile commonalities between the microbiota composition in AB-induced dysbiosis and that in ICI non-responders, specifically *Clostridium hathewayi* and *Clostridium clostridioforme*, which were ultimately proven to confer ICI treatment resistance. Moreover, ICI responders and non-responders each demonstrate distinct microbiome profiles highlighting specific intestinal microbiota commensals affecting clinical outcomes to ICIs. Routy et al. [13], through analysis of microbiome composition in RCC patients on ICIs, showed that the abundance *of Akkermansia muciniphilia* at the time of diagnosis correlated with a favorable response to treatment. Similarly, Salgia et al. [11] have reported increased *Akkermansia muciniphilia* relative abundance in responders which decreased in non-responders. Matson et al. [14] demonstrated the abundance of Bifidocacterium spp. in metastatic melanoma responders to anti-PD-1 targeted therapy. Sivan et al. [15] also reported mirroring results in a melanoma mouse model proving response to anti-PD-L1 with Bifidocacterium spp. abundance. Vetizou et al. [16] reported that Bacteroides spp. presence correlated with response to anti-CTLA-4 treatment in melanoma murine experimental models. Hence, the distinctive variation in microbiome profiles among different patients validates the relationship between gut microbiota and mRCC treatment outcomes and also the utility of using pre-immunotherapy microbiome fingerprints as potential predictors of ICI resistance and patient response to treatment. More importantly, it also raises novel insight into altering patients’ microbiome composition to increase ICI clinical efficacy through the use of alternative strategies such as fecal microbiota transplantation (NCT03772899), probiotics, or other nutritional supplements, such as curcumin.

In line with significant preclinical data extracted from studies performed on mice, evidence from human data shows that gut microbiota is an essential role player in influencing the response to cancer immunotherapy [16]. Alteration of the gut microbiome, either by antibiotics administration or by the usage of germ-free mice in experiments, has been associated with worse immunotherapy treatment outcomes [17]. Oral gavage with *B. fragilis* (*Bf.*), adoptive transfer of memory *Bf-*specific TH1 cells, and immunization with *Bf* polysaccharides demonstrated a positive response to anti-CTLA-4 in antibiotics-treated or germ-free mice [16]. This emphasizes the importance of the gut microbiome composition and the significant effect of its dysbiosis. Figure 1 depicts how gut dysbiosis leads to immunomodulation resulting in different treatment response.

The mechanism by which gut microbiota affects cancer immunotherapy is still not well established; however, studies have proposed three processes: stimulation of anti-tumor T cells directly by bacterial components, molecular mimicry between bacterial and tumoral epitopes, and anti-tumor immunity by bacterial metabolites [18]. Through the first mechanism, *Enterococcus hirae* and *Barnesiella intestinihominis* were shown to improve the efficacy of cyclophosphamide therapy. *E. hirae* translocates from the small intestine to secondary lymphoid organs and raises the intratumoral CD8/Treg ratio. Additionally, *B. intestinihominis* enhances the activity of IFN-γ–producing γ*δ* T cells [19]. Furthermore, improvement in the efficacy of CTLA-4 treatment was influenced by the presence of *B. thetaiotaomicron* and/or *B. fragilis* [16]. As for molecular mimicry, gut microbiota, specifically *Bifidobacterium breve* and *Enterococcus hirae,* stimulate commensal-specific T cells which cross-react with tumor-associated antigens [20]. For example, Fluckiger et al. reported the presence of major histocompatibility (MHC) class I-binding antigens in the tape measure protein (TMP) of enterococcal bacteriophage. The existence of these enterococcal bacteriophages in stools along with the tumor expression of TMP-cross-reactive epitopes was associated with increased overall survival in NSCLC and RCC patients subjected to anti-PD1 therapy [20]. Moreover, as for the last proposed mechanism, experiments performed on mice revealed that metabolites such as inosine produced by gut bacteria, specifically *A. muciniphila* and *B. pseudolongum*, improved response to targeted immunotherapy, anti-CTLA-4 in particular [21]. The suggested mechanisms emphasize the role of gut microbiome in immunotherapy and can serve as a basis for establishing microbial-based therapies.

Several studies, including Routy et al., have established a relationship between pretreatment gut microbiota and progression-free survival (PFS) and overall survival (OS) in patients with myeloma, hepatocellular carcinoma, and non-small cell lung cancer. Different bacterial species have been identified in responders versus non-responders (Table 1) [9,22,23]. In a review by Oh et al., it was identified that abundance of *Firmicutes* and *Actinobacteria*, specifically *Fecalibacterium prausnitzii* and *Bifidobacterium longum* was associated with favorable outcomes. It was postulated that the baseline gut microbiome could predict outcomes of immunotherapy [22].

Other studies have shown a correlation between metagenomics and therapy response. Differences of α diversity (high variation of microbes in a single sample) and β diversity (difference in diversity of microbes between sites) in responders compared with non-responders were highlighted. In patients with myeloma, prolonged PFS was seen with α diversity compared with intermediate and low diversity [22]. Zheng et al. also highlighted the role of α diversity in hepatocellular carcinoma (HCC) by observing the increase in *Proteobacteria* from week 3 to 12 in non-responders. The same study showed differences in β diversity across patients with HCC as early as week 6 of anti-PD1 therapy, further confirming the correlation between gut microbiome and clinical outcome following immunotherapy. Jin et al. reported a greater frequency of unique memory CD8^+^ T cell and natural killer cell subsets in the periphery in response to anti-PD-1 therapy, further documenting the importance of high diversity of microbiota in prolonging the PFS in patient with NSCLC on anti-PD1 immunotherapy [22,23]. Similar results were replicated by Giordan et al. when studying the antitumor effect of gut microbiota such as *Bifidobacterium* and *Faecalibacterium* species [22,23].

Similarly, relationships were reported between gut microbiota and incidence of gastrointestinal toxicity. It was noted that abundance of Bacteroidetes was associated with less incidence of immune therapy-related colitis by enhancing differentiation of T cells into T reg cells [22].

## 3. Proton Pump Inhibitors (PPIs) and Antibiotics’ Effect on the Immune Response

Gut bacteria are affected by medications. Since concomitant medication are often antibiotics and PPIs, it is safe to say that exposure to these medications could also be a factor in the equation. Routy et al. showed the effect of antibiotic use on PFS and OS in patients with non-small cell lung cancer (NSCLC), renal cell carcinoma (RCC), and urothelial carcinoma, further underlining the relation between dysbiosis and ICI response [24]. It is no secret that antibiotics and PPIs alter gut microbiota, and thus could be correlated to variation in the response to checkpoint inhibitors. In a study by Giordan et al. in 2021, it was shown that antibiotics and PPIs administered prior to starting immunotherapy impact prognosis negatively. Further, it was shown that the combination of antibiotics and PPIs could possibly have a deleterious synergistic effect [24]. The mechanism involved is not well understood. It was postulated that by causing dysbiosis, these medications alter the immunostimulatory effect of the microbiome, leading to bad outcomes. PPIs may affect the abundance and diversity of Bifidobacterium and Ruminococcaceae, both indispensable to immunity. Additionally, as the pH in the tumor microenvironment increases, PPIs decrease the intratumoral immunosuppressive activity and thus the activity of ICI [22] (Figure 2).

## 4. Dysbiosis and the Urothelial System

### 4.1. Discovering the Urinary Tract Microbiota

Conventional bacterial-dependent detection techniques established the long-held dogma of the proximal urinary tract as being ‘sterile’ [25]. It was speculated that the urine of a healthy subject is sterile, and in turn, identifying pathogens using standard bacterial culture techniques usually lead to the diagnosis of a UTI [26]. To date, the diagnosis of UTI remains ill-defined and ambiguous, creating room for unnecessary antibiotic overuse [27]. In the clinical setting, standard culture methods are mostly targeted to support the growth and propagation of specific organisms known to cause UTIs such as Escherichia coli, Staphylococcus saprophyticus, and Enterococcus faecalis, overlooking other slow-growing organisms such as Lactobacillus and Corynebacterium [28]. We now know, however, that there are immensely more bacterial populations inhabiting the urinary tract than that we can culture in the laboratory setting. More advanced techniques such as expanded quantitative urine culture (EQUC) and culture-independent detection techniques relying on molecular sequencing technologies such as 16S ribosomal RNA sequencing could detect bacterial genetic material [25,29]. These improved technologies, though still underestimating the microbial complexity, have identified more of the overlooked organisms as being part of what is now known as the microbiota of the urinary tract [25].

### 4.2. The Significance of the Urinary Microbiome in Disease

Ongoing studies of the microbial composition of the urinary microbiota have demonstrated significant differences between healthy patients and those with urological disease. Important microbial alterations were reported in patients with urological diseases, such as interstitial cystitis, urgency urinary incontinence, and kidney stones [30]. In patients with interstitial cystitis, urine studies demonstrated decreased bacterial diversity and increased Lactobacillus relative abundance compared with healthy controls [31]. Patients with urgency urinary incontinence were found to have a decreased urinary microbiome diversity as well with an increased abundance of Aerococcus and Gardnerella species [30,32]. Moreover, urinary microbiota dysbiosis was found to have a significant impact on the pathophysiology of prostate disease [33]. Yu et al. found differences in the microbial composition of expressed-prostatic-secretions (EPS) in patients with benign prostate hyperplasia (BPH) compared with those with prostate cancer, BPH patients having increased abundance of Eubacterium and Defluviicoccus genera and reduced abundance of Bacteroides and Firmicutes. These results suggested that chronic inflammation and proinflammatory cytokines could be propagated by the presence of specific bacteria and that the urinary microbiota has an important relevance in prostate disease [33]. Additionally, Bossa et al. concluded that UTI development was preceded by alterations in the urinary microbiome which were modulated after treatment [34]. Additionally, there exists a remarkable difference in the urinary microbiota composition between males and females owing to their anatomical and hormonal differences, which might explain differential disease susceptibility between the two sexes [35]. For example, Ghani et al. demonstrated an increased incidence of kidney stones in males compared with females [36].

### 4.3. The Urinary Microbiome and Bladder Cancer

Bladder cancer is one of the most common malignancies diagnosed, with an estimated incidence of 400,000 cases and more than 160,000 reported deaths per year. Its morbidity and mortality continue to increase due to external risk factors, such as tobacco smoke, and environmental exposures [37]. Apart from the genetic component, the etiology of bladder cancer remains multifaceted and unclear [38]. Current efforts on bladder cancer research have recognized the urinary microbiome as an important factor in the development of bladder cancer and its therapeutic response to treatement [39]. A study by Wu et al. has characterized the urinary microbiome profile in bladder cancer patients showing higher abundances of *Acinetobacter* and *Anaerococcus* genera compared with the non-cancer group [38]. Roperto et al. have also shown *Acinetobacter*’s abundance in the urine of cattle with bladder cancer [40]. It has also been reported that *Stretococcus* spp. and *Fusobacterium* genus were both more abundant in the bladder cancer group compared with the non-cancer group [38]. Xu et al. reported mirroring results with *Streptococcus* enrichment in the urine of urothelial carcinoma patients compared with healthy patients in which *Streptococcus* abundance was near zero [41]. Bucevic Popovic et al. also compared urine microbial profiles and found an increased abundance of the genus Fusobacterium in the bladder cancer group [42]. Immune checkpoint modulators, particularly PD-L1 inhibitors, are becoming a cornerstone in the management of local and metastatic urothelial carcinoma [39]. As mentioned in previous sections, more studies are emerging on the association between gut microbial composition and response to anti-PD-1/PDL1 [38]. A better understanding of the link between the urinary microbiota and bladder cancer could provide important directions for future research into targeting the urinary microbiota to enhance bladder cancer therapeutic response to immunotherapy [39].

### 4.4. Progression of Kidney Injury in Diabetic Nephropathy

Diabetic nephropathy (DN) is a common serious microvascular complication of diabetes mellitus (DM) and is associated with significantly increased morbidity and mortality [43]. Around one third of DM patients develop DN, which is a leading cause of end-stage renal disease (ESRD) [44]. Five stages (Stage I to V) of DN exist and are categorized based on glomerular filtration rate (GFR) and albuminuria [44]. Most of the time, DN is diagnosed once patients have already progressed to stage III or IV, which is considered irreversible [45]. Therefore, it is crucial to identify significant risk factors involved in the pathogenesis and progression of DN to prevent renal fibrosis, the final severe complication of DN [46].

The pathogenesis of DN is multifactorial, including renal hemodynamic changes, oxidative stress, hypoxia, inflammation, and overactivation of the renin–angiotensin–aldosterone system (RAAS) [47]. Several clinical studies have revealed that intensive glycemic control, inhibition of RAAS activation, and anti-inflammation have helped delay but not completely suppress the progression of DN [48]. Hence, it is necessary to identify other additional risk factors that play a fundamental role in DN progression.

Recent studies have revealed an association between gut microbiota and the kidney; it was shown that the dysbiosis in gut microbiota participated in DN progression [49]. In fact, the relation between gut microbiota and kidney constitutes a vicious cycle [49]. On one hand, the dysbiosis of the gut microbiota aggravates chronic inflammation further promoting kidney injury, and on the other hand, the resulting kidney injury contributes to gut milieu modification and eventually dysbiosis [46].

Furthermore, the gut bacterial dysbiosis triggers the release of lipopolysaccharides and the buildup of uremic toxins, further exacerbating kidney injury [46]. In addition to the latter, the gut bacterial community plays an important role in carbohydrate fermentation and the production of short chain fatty acids (SCFA), mainly acetate, propionate, and butyrate [50]. Other than the essential role of SCFAs in signaling and energy metabolism, SCFAs have been shown to promote immune response, regulate inflammatory reactions, and enhance insulin sensitivity, thus further lowering blood glucose level [50,51]. In DN patients, there is documented evidence of a notable decrease in the proportion of butyrate-producing bacteria with an increase in some opportunistic pathogens [52]. Butyrate is one of the most effective SCFAs implicated in DN [51]. Cai et al. reported a positive correlation of serum butyrate level with estimated glomerular filtration rate (eGFR) and a negative correlation with urine albumin-to-creatinine ratio (UACR) [52]. Additionally, butyrate increases the level of glucose transporter-4 (GLUT-4), the insulin-regulated glucose uptake transporter in skeletal muscle and adipose tissue, reducing serum glucose level [53]. Butyrate was also speculated to induce autophagy through AMP-activated protein kinase (AMPK)/mammalian target of rapamycin (mTOR) pathway, thus halting the progression of DN [52]. This suggests that butyrate with its diverse roles may serve as a potential therapy for prevention of DN progression [54]. To study the therapeutic role of butyrate, treatment modalities such as fecal microbial transplantation (FMT) were attempted; they have been shown to increase the level of fecal butyrate and alleviate renal injury. However, FMT was not cleared as safe enough to be used in clinical practice due to its imposed risk of severe bacteremia [51]. Additionally, several therapies such as probiotics and prebiotics intervene with intestinal flora composition and provide favorable effects in patients with DM by attenuating insulin resistance; however, there are also no conclusive results for their clinical application [48,50].

Moreover, multiple other factors can intervene in gut microbiota dysbiosis, eventually worsening DN progression. Diet is an essential contributor in which, for example, a potassium-rich diet can further decrease SCFA level and drive toward uremia [46,51]. In addition, medications such as anti-glycemic, anti-hypertensive, and anti-lipemic drugs modify gut microbiome and contribute to the pathogenesis of renal injury [49]. All the above findings, summarized in Figure 3, emphasize the relation between the gut and the kidney in hopes of establishing a promising therapy to arrest DN progression.

### 4.5. Effect of Dysbiosis on Kidney Stones That Leads up to Renal Cell Carcinoma (RCC)

Urolithiasis or kidney stone disease (KSD) is one of the most frequent urological pathologies, with its incidence reaching up to 20% worldwide [55]. It is a highly recurrent condition that affects all age groups and alters patients’ quality of life with a resulting economic burden [55,56]. The main initiator of KSD is the super saturation of urine with calcium and oxalate, causing the formation of stones [57]. The etiology of KSD is multifactorial, comprising genetic variations, dietary habits, socioeconomic status, antibiotics supplementation, and metabolic factors [57]. Recent studies have reported an association between microbiota in the gut and urinary tract and the development of kidney stones, and it was presumed that the dysbiosis in the latter microbiota contributed to the pathogenesis of KSD [58].

In KSD, dysbiosis in gut and urinary tract microbiome can be explained by the decrease in probiotic or favorable microbiota in addition to the abundance of pathogenic microbiota [59]. There is an evident difference in the diversity of microbiota in KSD patients compared with healthy controls [56]. In KSD patients, pro-inflammatory bacteria are the dominant microbes, compromising the gut barrier and modifying its permeability [59]. In fact, the reduction in the level of *F. prausnitziis* and *Bifidobacterium* in KSD patients results in decreased production of butyrate, a short-chain fatty acid (SCFA), leading to a marked inflammatory state, an environment favoring kidney stone formation [56]. In addition, the increase in the level of *Escherichia-Shigella* leads to the depletion of citrate levels and aggravates the development of kidney stones [56]. This dysbiosis and imbalance in the different populations of bacteria is a main factor in the pathogenesis of urolithiasis as seen in Figure 3 [60].

Moreover, several studies on KSD explored the role of *Oxalobacter formigenes*, a gram-negative anaerobic bacterium with oxalate-degrading property [58]. This property is due to the expression of two enzymes, Oxalyl coenzyme A decarboxylase and Formyl coenzyme A decarboxylase [59]. The human body lacks the ability to degrade oxalate; thus, bacteria are essential for oxalate metabolism and the prevention of oxalate stones formation [61]. *Oxalobacter* can reduce the excretion of oxalate in urine and can prompt the absorption of oxalate based on the oxalate gradient across the epithelium [56]. This was presumed to help further suppress the development of KSD [56]. In fact, the depletion of *Oxalobacter formigenes*, in the feces of KSD patients or in patients of high lithogenic risk, provided evidence to the potential contribution of this bacterium in preventing kidney stone formation [62]. However, Ticinesi et al. reported the isolation of *O. formigenes* from the feces of recurrent kidney stone formers; thus, the role of *O. formigenes* in lithogenesis has not been elusive [62]. Additionally, the probiotic administration of *Oxalobacter* or other oxalate-degrading species did not significantly decrease the excretion of oxalate in urine even though the bacterium was isolated from feces of subjected patients as proof of its presence in gut microbiota [56,58]. Results from studies on *O. formigenes* are still controversial, and no clear consensus on the correlation between this bacterium and KSD has been established [58].

As stated previously, dysbiosis and imbalance in the gut and urinary tract microbiome is a key factor in KSD that leads to a higher incidence of kidney stones occurrence [56]. The latter is concerning because there has been a strong association between kidney stones and kidney cancers [56]. The prevalence of both renal cell carcinoma (RCC), the most common type of renal parenchymal tumor and transitional cell carcinoma (TCC), the malignancy of the upper urinary tract involving the renal pelvis and ureter, has been gradually increasing over the years [56]. Of importance, several studies revealed that chronic inflammation and infection resulting from KSD could alter the proliferation of urothelial cells, leading to tumor development [63]. According to a study carried out by Chow et al., kidney stone patients were at a higher risk of developing renal pelvis, bladder, or ureter malignancies because of marked irritation and infection since kidney stones form in the same location where tumors originate [63]. Similarly, the Netherlands Cohort Study (NLCS) demonstrated that compared with patients with no history of kidney stones, those with kidney stones were at an increased risk of developing TCC and papillary RCC but not clear-cell RCC [63]. Additionally, diagnosis of kidney stones before the age of 40 is associated with a higher risk of RCC and TCC than diagnosis at a later age [63]. Nonetheless, the association between kidney stones and malignancy can be ascribed to their common risk factors such as hypertension, DM, smoking, obesity, and dietary factors [56]. Accordingly, it is important to highlight significant factors in the pathogenesis of kidney stones in aim of preventing malignancy.

To prevent kidney stones and hence kidney cancers, several promoting risk factors can be controlled such as diet [58]. Nutritional imbalances associated with high lithogenic risk include diets high in salt, animal protein, and oxalate, and low in calcium, fruit, and vegetables, as well as poor hydration [58]. The high concentration of salt leads to gut microbial dysbiosis. It leads to a depletion of *Lactobacillus*, *Akkermansia*, and *Bifidobacterium*, resulting in increased urine calcium and decreased urine citrate, further elevating the risk of nephrolithiasis [56,58]. Additionally, the high consumption of animal protein promotes the increase in pathologic bacteria such as *Escherichia-Shigella* and the depletion of SCFA-producing bacteria such as *Faecalibacterium*; this imbalance in microbiota raises the risk for kidney stone formation [56,58]. On the other hand, low calcium, fruit, and vegetables and poor hydration contribute to the depletion of SCFA-producing, lactic acid-producing, and oxalate-degrading species, which causes a decrease in urinary volume and an increase in urine calcium and oxalate levels [56,58]. In short, controlling dietary habits is considered a possible nonpharmacologic management for dysbiosis intended to halt the development of kidney stones and the consequent possible cancer [56,58].

### 4.6. The Vaginal Microbiota

Notwithstanding the myriad studies exploring the gut microbiota, the vagina is another high-volume microbiota organ. The vaginal microbiota comprises around 9% of the total human microbiota and is a central component of reproductive health and disease [64,65]. For decades, the normal vaginal flora has been thought to comprise predominantly lactobacilli; however, more advanced non-culture-based modern techniques have exposed a more diverse composition of the healthy vaginal flora across females of different origins, which contains more than 50 non-pathogenic organisms. This flora is a dynamic microenvironment that is affected by a female’s menstrual cycle, sexual activity, gestational status, and contraceptive use [65]. Additionally, studies have shown that certain vaginal microbiota profiles dictate genitourinary health and disease [66]. For example, the vagina is a potential niche for pathogenic E. coli colonization, and studies have found E. coli isolates in the vaginal introituses of women who suffer recurrent UTIs compared with healthy controls [67]. Moreover, urinary E. coli were found to invade vaginal cells and colonize the vagina after a UTI, seeding back into the urinary bladder to trigger ascending infections and recurrent UTIs [68]. Studies have highlighted the protective function of vaginal lactobacilli with lactobacilli-dominant microbiota conferring less risk for UTI development [69]. Emerging studies are using probiotics tackling vaginal lactobacilli to decrease the incidence of post-op UTIs [70]. The gut, vaginal, and urinary microbiomes have proved to be a joint trifecta implicated in both health and disease states and need to be profiled and characterized to exploit novel findings.

## 5. Novel Therapeutic Approaches in Management of Urological Disease

Given the gut microbiota’s role in UTI development and prevention, different studies were directed to modulate its composition as a novel strategy of management. In a randomized, double-blinded pilot study, Koradia et al. concluded that supplementation with lactobacilli probiotics and cranberry extract decreased UTI recurrence in a population of premenopausal women compared with control [71]. Tariq et al. stated that fecal microbiota transplantation performed for the management of *C. difficile* infection also decreased UTI occurrence, which was also reported in a study by Wang et al. [72,73]. Newer studies are emerging on the modulation of the urinary microbiota as an approach in management of urological disease. Murphy et al. demonstrated a potential intervention in the treatment of chronic prostatitis and chronic pelvic pain syndrome by performing intraurethral instillation of the commensal *Staphylococcus epidermidis* isolated from prostate secretions in healthy controls. It was reported that this intervention could decrease prostatitis-related symptoms [74].

### 5.1. Curcumin

#### 5.1.1. Curcumin’s Anti-Inflammatory Effects and the Gut Microbiota

Given the growing body of evidence that the human gut microbiota could represent a determinant in disease development and progression, research has been instituted to study its connections with different inflammatory and non-inflammatory conditions. In fact, gut microbial imbalances have been observed in several inflammatory conditions such as chronic kidney disease (CKD) and obesity [75,76]. One possible strategy to control disease outcomes is to alter the gut microbial profile through the use of dietary supplements, among which curcumin has been most studied [75]. Curcumin (CUR) is a natural polyphenolic compound and the active ingredient in the Indian dietary spice turmeric (*Curcuma longa*), which also holds the curcuminoids demethoxicurcumin and bisdemetoxicurcumin [77]. Curcumin has long been studied for its anti-inflammatory, anti-oxidative, anti-microbial, and anti-proliferative potential, and its supplementation has been proven to significantly alter the gut microbiota composition [75]. Through its ability to increase the expression of intestinal alkaline phosphatase, decrease intestinal permeability, and decrease the production of inflammatory cytokines, curcumin can potentially change the gut’s microbial communities [78]. Several studies and clinical trials (NCT04413266) are on the rise to classify curcumin as a promising adjunct in the treatment of CKD and its complications [75,77]. Pivari et al., through gut microbiota genome sequencing and analysis, found a significant shift to Lachnospiraceae, Prevotellaceae, and Ruminococcaceae family abundance in CKD patients supplemented with curcumin for six months, concluding that the latter’s gut composition shifted towards that of the healthy controls [75]. In addition, curcumin’s anti-inflammatory properties have been reported in multiple in-vitro studies. Pivari et al. proved that curcumin’s anti-inflammatory action is attributed to a decrease in pro-inflammatory mediators (CCL-2, IFN-gamma, and IL-4) [75], and Alvarenga et al. also reported a decrease in inflammatory cytokines by direct inhibition of nuclear factor kappa-B (NF-kB) [79].

#### 5.1.2. Curcumin, an ‘Immunotherapy Supplement’

Similarly, recent studies have highlighted curcumin’s anti-proliferative and hence anti-tumorigenic effects through its actions on different carcinogenic biochemical and signaling pathways [7]. It has been reported to inhibit cell cycle proliferation, tumor invasion, and angiogenesis and to promote apoptosis through induction of anti-apoptotic proteins in different cancers, particularly prostate, breast, and colon cancer [80,81]. These properties have also been demonstrated in multiple clinical trials in patients with prostate cancer (NCT02724618), pancreatic cancer (NCT02724202), and other cancers [81]. Newer studies are on the rise showing the effects of curcumin on signaling pathways implicated in RCC progression. Gong et al. [7] reported that it suppresses the AKT/mTOR pathway, leading to apoptosis and autophagy in RCC. In addition, Xu et al. demonstrated that curcumin inhibited the viability of clear cell renal cell carcinoma cells through the NF-kB and AKT signaling pathways. Thus, with the deeper understanding of the molecular biology and biochemical pathways of RCC, curcumin proves to have a direct role in modulating RCC oncogenesis, and hence its use has been reported [82]. Contrary to CKD, meager data are available on the causality rather than just the association between the gut microbiota and mRCC oncogenesis. However, and as discussed above, select commensals have been implicated in ICI response and non-response [9,11], and translating this rather equivocal knowledge into using microbiota modulators in mRCC management is not an easy task. Given the notable effect curcumin has on the gut microbiome composition, further endeavors must be directed to analyze a bigger mRCC patient population on ICIs with adjuvant curcumin supplementation. This combination can change the therapeutic landscape of mRCC, making curcumin an accessible, non-toxic, dietary ‘immunotherapy supplement’ with immense potential in oncology.

### 5.2. Effect of Fecal Microbiome Transplant

Recent fecal microbiome transplant (FMT) studies highlighted the effects of FMT on changing the composition and diversity of the gut microbiota in treatment non-responders. In myeloma, Baruch et al. showed a response in treatment refractory patients after FMT from responders and reintroduction of nivolumab (anti-PD1 immunotherapy) [83]. The safety and feasibility of this practice were also highlighted. A similar study by Davar et al. also showed the modulatory effect of FMT in melanoma patients on pembrolizumab [84]. Additionally, several clinical trials were conducted to investigate the role of FMT in cancer therapy (Table 2).

The role and the mechanism of gut microbiota effect on immune therapy response and outcome is not well understood. It is postulated that dysbiosis could be associated with host genetic mutations leading to alteration in the immune system. Similarly, antibiotics and PPIs might alter the patient immunity, leading to differences in response to therapy. It is known that Treg cells suppress lymphocytes. This became relevant when some favorable bacteria were found to be associated with decreased peripheral Treg levels, thereby halting the suppression of the lymphocyte. These low levels of Treg seemed to correlate with good response to immunotherapy, further drawing an association between bacteria and response to treatment [85]. The communication between gut microbiota and immunological response could possibly be mediated by products of bacteria metabolism [85]. It is no secret that gut organisms are associated with chronic inflammation and release of cytokines and chemokines. Thus, an enhanced infiltration of dendritic cells, TH1 cells, and CD8+ in the tumor microenvironment is, to date, the leading mechanism of bacterial immunomodulation [23].

## 6. Conclusions

While there is emerging evidence on the microbiome’s relevance in urological health, the extent of this association is still not compelling. There is still a need for considerable research to characterize the microbiome in the urinary tract and establish specific causal relationships among bacterial genera and associated conditions. The impact of this new paradigm encompasses not only the pathophysiology of urological disease but also shifts interest into a novel therapeutic armamentarium. The introduction of microbiome-targeting interventions, such as probiotics, to restore urological eubiosis could reduce antibiotic overuse and improve patient outcome. 

## Figures and Tables

**Figure 1 cancers-14-03328-f001:**
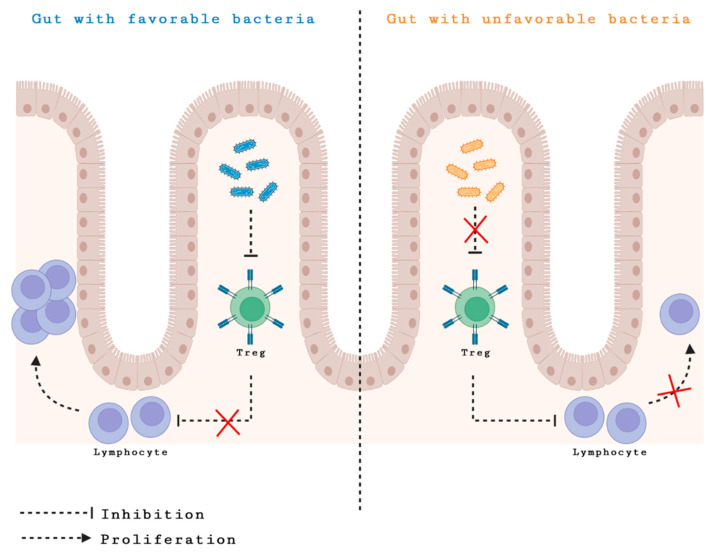
Gut dysbiosis leading to immunomodulation resulting in different treatment response.

**Figure 2 cancers-14-03328-f002:**
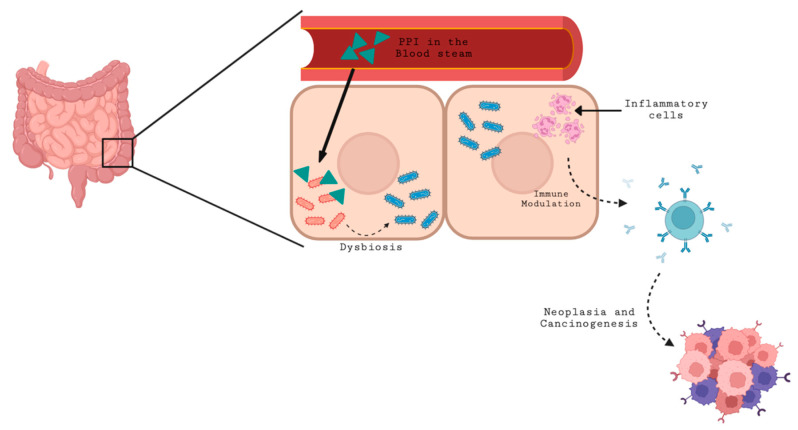
Effect of PPI on gut microbiota and oncogenesis.

**Figure 3 cancers-14-03328-f003:**
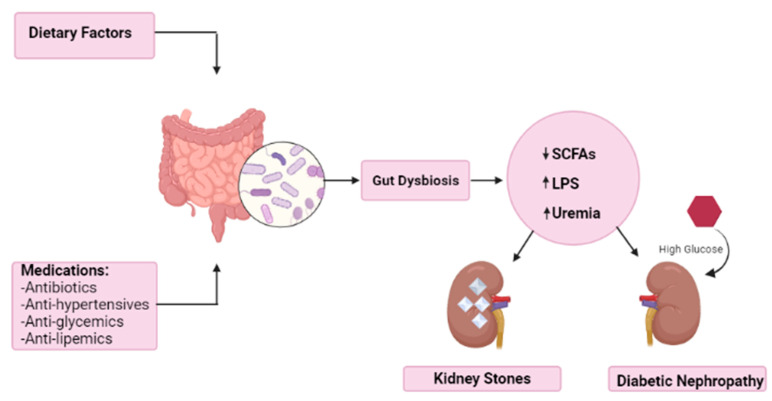
The effect of diet and medications on gut dysbiosis leading to kidney stones and diabetic nephropathy. (SCFA = Short Chain Fatty Acid, LPS = Lipopolysaccharide).

**Table 1 cancers-14-03328-t001:** Difference in gut microbiota between responders and non-responders.

	Responders	Non-Responders
Abundance	*Fecalibacterium, Bacteroides, Holdemania, Gemmiger*, and *Clostridium* XIVa, *Ruminococcaceae, Bifidobacterium longum*, *Collinsella aerofaciens*, *Enterococcus faecium, Coprococcus eutactus*, *Prevotella stercorea*, *Streptococcus sanguinis*, *Streptococcus anginosus*, *Lachnospiraceae bac- terium, Akkermansia muciniphila*	*Ruminococcus obeum, Roseburia intestinalis, Bacteroides ovatus*, *Bacteroides dorei*, *Bacteroides massiliensis*, *Ruminococcus gnavus*, and *Blautia producta*
Non-Abundance	*Bacteroides*, *Lachnospiraceae, Clostridium IV, Blautia, Eubacterium*	

**Table 2 cancers-14-03328-t002:** This table lists clinical trials investigating the role of FMT in cancer therapy. (dMMR: Deficient, Mismatch Repair, FMT: Fecal Microbial Transplant, PD-1: Programmed Cell Death-1).

Clinical Trial Identifier /Sponsor	Source of FMT	Intervention	Type of Cancer	Antibiotics	Recruitment Status	Objective Response Rate	Long Term Clinical Benefit	Adverse Events Related to Treatment
Davar et al. [84]	Melanoma PD-1 responder	200 mg IV Pembrolizumab over 30 min on Day 1 of cycle (same day as the FMT). Total of 3 cycles done.	PD-1 secondary refractory melanoma	Not given	Active, not recruiting	20%	40% of patients with advanced melanoma	Grade 3 treatment related adverse events (2 cases of fatigue and 1 peripheral motor neuropathy), no grade 4/5 adverse events reported
Baruch et al. [83]	Melanoma PD-1 responder	FMT via colonoscopy (protocol day 0) then FMT packed into capsules given (Day 1 and 12), repeated every 2 weeks along withNivolumab 240 mg.	PD-1 primary and secondary refractory melanoma	Pre-FMT vancomycin and neomycin	Unknown	30%	Not reported	No moderate to severe treatment-related adverse event (grade 2–4)
NCT04729322	dMMR PD-1 responder	FMT via colonoscopy (Day 5 of cycle 1) then FMT capsules on days 1, 8 and 15. FMT capsules given on day 1 every 3 weeks for cycles 2 and onward.	Metastatic colon cancer	Pre-FMT Metronidazole, vancomycin, neomycin	Active, recruiting	Not reported	Not reported	Not reported
NCT04130763	Donors with gut microbiota profile similar to PD-1 responders	FMT capsules for 1 week for cycle 1 as induction. FMT capsules for cycles 2 and onward as maintenance.	GI cancer after failure of anti-PD-1 treatment	Not given	Active, recruiting	Not reported	Not reported	Not reported
NCT03772899	Healthy donor selected via protocol	FMT at least one week prior to treatment with either immunotherapy followed by FMT along with Nivolumab or Pembrolizumab as maintenance	Unresectable or metastatic cutaneous melanoma (BRAF wild type or mutant)	Not given	Active, not recruiting	Not reported	Not reported	Not reported

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
