# Peer review of "The Urogenital System’s Role in Diseases: A Synopsis"

_cancers, 2022, doi:10.3390/cancers14143328_

Round 1
Reviewer 1 Report
Zerdan et al. have attempted to review the microbiome in the urogenital system and its relationship to the gut microbiome. Whilst the majority of the sections covered in the paper are well-written, the structure of the review is muddled/confusing and there are several issues which I feel need to be addressed:
1. The Summary section of the paper does not seem to represent the true content of the review article. In fact when reading this summary I thought the review article was going to focus mainly on cancers of the urogenital system yet it doesn’t?
2. The structure of the review article is also slightly confusing/ mixed as the final section 7 is on Novel Therapeutic Approaches in Management of Urological Disease yet the effect of fecal microbiome transplant was dealt with in section 5.
3. Section 7.2 seemed to overly focus on the direct role of curcumin in various cancers which was detracting from the focus of the paper
4. The title of the review article specifically states the gut microbiome and the urogenital system: a growing relationship, yet Section 6 deals with the urinary tract microbiota with no relationship to the gut microbiota.
5. In section 6.2 entitled The Significance of the Urinary Microbiome in Disease no mention was made of the role of the urinary microbiome in bladder cancer of which there is a significant body of published work.
6. In Section3 Human gut microbiota composition and RCC oncogenesis, it should be made clear when referring to references whether the study was done in RCC or in fact in other cancer types such as ref 18 which was in metastatic melanoma patients, ref 19 which was in melanoma mouse model, ref 20 which used a sarcoma mouse model and malignant melanoma patients.
7. No mention was made of the dynamic shift in microbiome that occurs across the female lifecycle and how it contributes to maintaining vaginal health. Bacterial strains resident in the gut and vagina crosstalk, which leads to local and systemic immune regulation. The authors made no reference to understanding the connection between intestinal and vaginal microbiota which may represent an avenue for new treatments of female genital tract disorders.
Author Response
1. The Summary section of the paper does not seem to represent the true content of the review article. In fact when reading this summary I thought the review article was going to focus mainly on cancers of the urogenital system yet it doesn’t?
—> the summary has been taken out. We agree that it doesn't represent the topics discussed in this manuscript.
- The structure of the review article is also slightly confusing/ mixed as the final section 7 is on Novel Therapeutic Approaches in Management of Urological Disease yet the effect of fecal microbiome transplant was dealt with in section 5.
—> Section 5 has been moved to section 7. This would offset the important of curcumin in that section.
- Section 7.2 seemed to overly focus on the direct role of curcumin in various cancers which was detracting from the focus of the paper
- The title of the review article specifically states the gut microbiome and the urogenital system: a growing relationship, yet Section 6 deals with the urinary tract microbiota with no relationship to the gut microbiota.
—> The role of the gut microbiota has been mentioned in other aspects of the manuscripts. if you believe section 6 should be removed or trimmed down, we wold be more than glad to do it. We also changed the title of the manuscript.
- In section 6.2 entitled The Significance of the Urinary Microbiome in Disease no mention was made of the role of the urinary microbiome in bladder cancer of which there is a significant body of published work.
—> This has been modified and it has been added.
- In Section3 Human gut microbiota composition and RCC oncogenesis, it should be made clear when referring to references whether the study was done in RCC or in fact in other cancer types such as ref 18 which was in metastatic melanoma patients, ref 19 which was in melanoma mouse model, ref 20 which used a sarcoma mouse model and malignant melanoma patients.
—> the references has been fixed. Irrelevant sources and data have been removed.
- No mention was made of the dynamic shift in microbiome that occurs across the female lifecycle and how it contributes to maintaining vaginal health. Bacterial strains resident in the gut and vagina crosstalk, which leads to local and systemic immune regulation. The authors made no reference to understanding the connection between intestinal and vaginal microbiota which may represent an avenue for new treatments of female genital tract disorders.
—> A section has been added.
—>

Reviewer 2 Report
In this paper, Bouzerdan and colleagues depict the current evidence concerning the interaction between the gut microbiome and several urinary tract disorders.
The review is well written and topics are clear. The table and figures are well done.
It is worthy of publication, some minor edits:
-The section about immunotherapy and gut microbiome should be elaborated further with more evidence.
-It is crucial to mention trials investigating probiotic supplements, FMT,... in cancer treatment response. One example would be PMID 35228755.
-It would be valuable to mention current evidence of the relationship between gut microbiome and response to targeted therapies.
-It would be enriching to add a figure about the possible interactions of the gut microbiome and renal oncogenesis.
Author Response
Thanks a lot for your positive feedback.
We have tackled all comments provided.
-The section about immunotherapy and gut microbiome should be elaborated further with more evidence.
—> this has been done and it is evident in the document attached (with tracked changes)
-It is crucial to mention trials investigating probiotic supplements, FMT,... in cancer treatment response. One example would be PMID 35228755.
These have been added as well.
-It would be valuable to mention current evidence of the relationship between gut microbiome and response to targeted therapies.
—> this has been added towards the middle section of the paragraph.
-It would be enriching to add a figure about the possible interactions of the gut microbiome and renal oncogenesis.
—> two figures have been added. We believe it is hard to do a better figure on such a short notice of time. We apologize for that. Please let us know if you would like to keep the figure or remove it.

Round 2
Reviewer 1 Report
Text needs to be checked for correct English and spelling errors as below:
In the Summary the following sentence should read: For that reason, tackling the urinary microbiome in healthy individuals, as well as its involvement in disease through the proliferation of opportunistic pathogens, could open a potential field of study, leading to new insights into prevention, diagnosis, and treatment strategies for different diseases.
Line 143 – spelling error
Matson et al. (18) have demonstrated the abundance of Bifidobacterium spp. in(14) in metastatic melanoma responders to anti-PD-1 targeted therapy.
Same spelling error for Bifidobacterium on line 146
Line 165 – poor English
Should read: This emphasizes the importance of the gut microbiome composition…
Line 195 should read: patients with myeloma, hepatocellular carcinoma, and non-small cell lung cancer.
Lines 204 and 205: sentence does not make sense: In patients with myeloma, prolonged PFS was seen with higher? α diversity compared to intermediate and low diversity
Line 312: spelling error: treatment
Line 316: spelling error: Streptococcus spp
Line 423: spelling error: F. prausnitzii
Author Response
Dear reviewer,
Thanks a lot for pointing out all those spelling mistakes we had. All suggestions were tackled. (a few more as well).